# Ensemble of Heterogeneous Base Classifiers for Human Gait Recognition

**DOI:** 10.3390/s23010508

**Published:** 2023-01-02

**Authors:** Marcin Derlatka, Marta Borowska

**Affiliations:** Institute of Biomedical Engineering, Faculty of Mechanical Engineering, Bialystok University of Technology, 15-351 Bialystok, Poland

**Keywords:** biometrics, human gait recognition, ground reaction forces, ensemble classifiers, classification, entropy

## Abstract

Human gait recognition is one of the most interesting issues within the subject of behavioral biometrics. The most significant problems connected with the practical application of biometric systems include their accuracy as well as the speed at which they operate, understood both as the time needed to recognize a particular person as well as the time necessary to create and train a biometric system. The present study made use of an ensemble of heterogeneous base classifiers to address these issues. A Heterogeneous ensemble is a group of classification models trained using various algorithms and combined to output an effective recognition A group of parameters identified on the basis of ground reaction forces was accepted as input signals. The proposed solution was tested on a sample of 322 people (5980 gait cycles). Results concerning the accuracy of recognition (meaning the Correct Classification Rate quality at 99.65%), as well as operation time (meaning the time of model construction at <12.5 min and the time needed to recognize a person at <0.1 s), should be considered as very good and exceed in quality other methods so far described in the literature.

## 1. Introduction

Biometrics, understood as the ability to recognize a particular person through the measuring of his or her physical characteristics or behaviors, is encroaching more and more boldly on our present reality. To meet the expectations of recipients, biometric systems should possess two basic characteristics:Security, seen both as high quality of recognition as well as resistance to unauthorized access attempts;System operation time, understood both as the time needed to recognize a given person as well as the time of model creation.

Generally, it can be claimed that static biometric systems based on the measurement of such features as fingerprints [1,2], retina [3], face [4], or hand geometry [5] ensure a relatively high level of people recognition. A drawback of such systems is that they are susceptible to activities allowing resource access to people who are unauthorized to do so. Dynamic systems based on measuring human behaviors, on the other hand, such as voice [6], handwriting [7], keystroke dynamics [8] or gait [9,10] recognition, although characterized by lower accuracy, are either very difficult or impossible to deceive. This is the reason that multimodal biometric systems using two or more biometric solutions of both types are often recommended [11,12].

To improve the quality of those systems authors focus on the identification of new attributes describing particular biometrics, methods of preliminary processing of data, as well as algorithms ensuring an improved level of accurate recognition [13,14,15]. Among the many algorithms used two approaches seem to be the most promising. The first is connected with deep learning, which through the use of multi-level neuron networks allows the achievement of previously unseen levels of accurate recognition [16,17,18]. One flaw of this approach is the need for many data as well as significant calculating power. The other approach is the application of a group of simple classifiers which share in determining the classifying decision [19,20,21]. The contrast with deep learning classifier ensembles, in general, consists of simple classifiers with less enhanced algorithms. Its main strength comes from the number of applied basic classifiers. Classifier ensembles may consist of similar (homogenous) or differing (heterogenous) base classifiers [22,23,24,25,26].

Most recent scientific works connected to human biometrics place a lot of emphasis on the operating time of biometric systems [27,28,29]. In commercial applications operating within real environments, a biometric system should permit both the addition of new users to the set of people who already have access to resources, as well as the removal of those who are no longer authorized to use them. This means that it not only needs to have a short reaction time, but an equally short scaling time as well.

From among behavioral biometrics, human gait recognition is especially intriguing. It is one of the most complex human activities and is performed unconsciously. It is believed that after becoming an adult a person’s gait barely changes. The human gait is also one of the few biometrics whose measurement, with an appropriately constructed testing path, does not require any interaction with the test subject. Furthermore, when cameras are used to recording human gait these measurements can be performed at a certain distance, with the test subject not being aware of the process. Human gait is practically impossible to duplicate.

### 1.1. Motivation

At this time, most works related to the biometrics of human gait focus on data recorded by cameras. Published study results mainly concentrate on the quality of biometric systems with the exclusion of model creation and recognition times. Among methods of classification, deep neural networks or ensembles of homogeneous classifiers are most often used. The present work aims to eliminate gaps in the literature related to the subject of human gait recognition with its main goal being the presentation of a method for the recognition of a particular person through the way he or she moves, using Ground Reaction Forces (GRFs) and an ensemble of heterogenous base classifiers. Additionally, it also addresses model creation as well as recognition times.

### 1.2. Contributions

The main contributions of this work are specified below:Human gait, a phenomenon that is cyclical and time-dependent, was described using synthetic features. These characteristics include various parameters starting with average waveform values through spectral entropy, and including Hjorth parameters to wavelet decomposition coefficients. Some of these have never been used before to identify a person through their gait;Classification results of selected base classifiers for particular groups of features were presented;The effectiveness of homogeneous and heterogeneous multiple classifiers, both with relation to classification quality and model creation and recognition times, were compared;The proposed human recognition algorithm was tested on a large sample of data gathered by the present work’s first author with this set being one of the most abundant databases reported within the literature.

## 2. Related Works

The recognition of a person through his/her gait was addressed for the first time in [30]. Within this work, it has been shown that a person can identify another person whom he or she knows by the way that person moves, even when their clothing or hairstyle is not typical, at great distances.

In the work [31] approaches connected to human gait recognition were grouped depending on the type of sensors used to gather data. The main methods of recognition included those relying on:Video cameras;Accelerometers and other wearable devices;The measurement of pressure exerted by a person’s foot on the ground.

For signals recorded using video sensors, Gait Energy Image (GEI) representation has been successfully employed. GEI is obtained through a simple average of silhouettes during walking. Modifications of this method which improve GEI effectiveness [32] are also utilized. Very good results were also obtained using other methods. In [33], Kinect was employed to acquire three-dimensional coordinates of human bones. Distances between bone nodes became the features, while the classifier was a support vector machine (SVM) using one-versus-one and one-versus-all algorithms to solve the multiclassification task. This approach yielded 99.8% correct identifications using data from 50 people. Another often-used approach is the application of deep Convolutional Neural Networks (CNN). Special deep CNN architecture was developed by [34] consisting of eight layers: four convolution layers and four pooling layers. This architecture is less sensitive to several typical variations and occlusions reducing the quality of gait recognition. The training time of this network for data contained within the CASIA-B database (124 people; less than 800 patterns) exceeded 9 min while the recognition time was 0.01 s. It is also worth mentioning that deep CNN is successfully used to classify various-sourced images as exemplified in the work of [35]. Article [36] presents the application of the Vision Transformer with an attention mechanism for gait recognition—or the GaitViT method. This instrument was tested on well-known databases, reaching 99.93% of correct classifications for the CASIA-B set, 100% for the OU-ISIR D, and 99.51% for the OU-LP databases.

The methods using accelerometers or other wearable devices are the most often used on mobile devices for authentication purposes. In [37] is presented the GaitPrivacyOn solution, which consists of two modules. The first one consisting of two autoencoders is transforming the raw data into privacy-preserving representation such as gender or activity. The second module is a mobile gait verification system CNN and Recurrent Neural Networks (RNN) with a Siamese architecture. The model was tested on popular databases MotionSense, MobiAct or OU-ISIR obtaining good classification results (more than 0.996 AUC). In the paper [38], to increase the quality of recognition the following methods were proposed: gait cycle segmentation with velocity adaptation and individualized matching threshold generation. The obtained results of the average gait recognition and user authentication rates are 96.9% and 91.75%, respectively.

Floor sensor methods measure the pressure exerted by the foot onto a sensor, usually hidden within the testing path. This approach allows the assessment of such values as ground reaction force [39], two dimensional center of pressure trajectories [40], or the time in which the foot remains in contact with the ground [41]. The lower popularity of FS methods undoubtedly results from their range of potential application being limited to situations where it is possible to force a person to walk through a testing path such as security gates at airports, supermarkets, workplaces or other buildings.

The number of articles addressing the problem of ensemble classifiers and time of classification in human gait recognition based on floor sensors field is very small. The work of [42] utilizes the fusion of time and spatial holistic pressure data gathered from 127 people recorded using a piezoelectric sensor mat. Better results were obtained through the combining of decisions of individual classifiers than through the application of feature-level fusion before decision made by a single classifier. The work of [39], on the other hand, presents a system made up of a multi-stage classifier set trained on the basis of different characteristic vectors for footstep patterns recorded from a special sensor material that covered an area of 100 m^2^. Unfortunately, this study was carried out using data from only 21 people. In the work of [43], in turn, a homogeneous ensemble of classifiers is presented that made decisions based on distances between components of ground reaction forces of the person being tested, in comparison to a pattern from a database. Distances were determined through the application of a dynamic time warping (DTW) algorithm with kNN classifiers as the base classifiers. In a study conducted on a sample of 99 women, 97.74% accurate identifications were achieved. However, the utilization of DTW requires that the database contains recorded gait cycles and the determination of distances, which is a relatively time-consuming task that occurs after the test of the person undergoing the procedure of recognition. Since the speed of homogenous multiple classifiers using kNN as base classifiers is closely connected to the number of patterns contained within the database, the entire process took nearly an hour. However, as has been shown by the work of [22], a larger database allows greater accuracy of recognition. In this situation, it becomes necessary to find a reasonable compromise between the accuracy and operational speed of a biometric system.

## 3. Materials and Methods

The overall flowchart appears in Figure 1. First, we measured GRFs by means of two force plates. Next, the features were calculated and divided into 7 groups. Each feature’s group has its own base classifier. Then, the decision of all of classifiers was made according to an assumed ensemble strategy. The result was the subject ID.

### 3.1. Measured Data

The force generated during walking between the foot and the ground is called the ground reaction force or GRF. To measure this force, plates made by the Kistler Company (Winterthur, Switzerland) utilize four piezoelectric sensors located in the corners of the platform. The signal measured by the sensors is employed to represent three components of GRF: anterior–posterior Fx, vertical Fy and lateral Fz (Figure 2).

Maximum values for the vertical component Fy correspond to the moments of transferring the entire body weight onto the analyzed limb (first maximum–maximum of the overload phase) and the load of the forefoot (the heel is not in contact with the ground) right before the toes off (the second maximum–maximum of propulsion). The anterior/posterior component Fx has two phases. In the first one its value is negative. This is the result of deceleration of the registered lower limb, because the direction of the force is opposite to the direction of walking. The minimum of the inhibiting phase is usually reached just before the peak force during the weight acceptance occurs for vertical component Fy. Analogically, in the second phase the anterior/posterior component takes positive values. Then begins the process of acceleration completed with taking the toes off the ground. The direction of the component Fz depends on the examined lower limb. Usually, it is assumed that values of Fz are negative for the left lower limb and positive for the right lower limb. Slight exceptions are the moments of initial contact and toe-off, where the foot is at a slight supination. The value of force Fz depends on the style of putting feet on the ground by the person under examination.

Measurements made as part of this study were performed using two Kistler platforms with the dimensions of 60 cm × 40 cm registering data with a frequency of 960 Hz. Ground reaction forces registered through the use of plates made by the Kistler Company form a time series *x*_1_, *x*_2_, …, *x_N_*, where *N* is the number of samples. Generally, the duration time of the support phase of a person’s gait depends on several factors and varies so *N* is variable.

### 3.2. Features

To eliminate the impact of the duration of the support phase on the possibilities of comparing two gait cycles, the following parameters were applied. It is also necessary to specify that signal features were calculated independently for each element of GRF and separately for each leg. The MNE features an open-source Python module for the extraction of features from signals [44], a pyeeg module with many functions for time series analysis [45], and an AntroPy module with several algorithms for computing the complexity of signals. Those modules include numerous parameters describing medical signals. Among algorithms implemented in modules the following were used:Mean of the signal is the sum of its elements:
(1)$$\bar{x}=\frac{1}{N}\sum_{i=1}^{N}x_{i}=\frac{x_{1}+x_{2}+\cdots +x_{N}}{N}$$

Variance of the signal is the average of the squared deviations from the mean:


(2)
$$var=\frac{1}{N}\sum_{i=1}^{N}{\left|x_{i}-\bar{x}\right|}^{2}$$


Standard deviation of the signal is the square root of the average of the squared deviations from the mean:


(3)
$$\sigma_{x}=\sqrt{\frac{1}{N}\overset{N}{\underset{i=1}{\sum }}{\left(x_{i}-\bar{x}\right)}^{2}}$$


Peak-to-peak (*ptp*) amplitude of the signal is the range of the signal:


(4)
ptp=(max(x)−min(x))


Skewness of the signal is computed as the Fisher-Pearson coefficient of skewness:(5)$$skew=\frac{m_{3}}{m_{2}^{3/2}}$$
where $m_{i}=\frac{1}{n}\overset{n}{\underset{i=1}{\sum }}{\left(x_{i}-\bar{x}\right)}^{k}$ is the biased *k*th sample central moment.

Kurtosis of the signal is the fourth central moment divided by the square of the variance:


(6)
$$kurtosis=\frac{m_{4}}{var^{2}}$$


Hurst exponent of the signal is calculated from the rescaled range and average over all the partial time series of length *N*:(7)$${\left(\frac{R}{S}\right)}_{t}=\frac{R_{t}}{S_{t}}$$
where *R/S* is averaged over the regions $\left[x_{1},\;x_{t}],\;\;[x_{t+1},\;x_{2t}]\;\;\mathrm{until}\;\;[x_{\left(l-1\right)t+1},\;x_{lt}\right]$ where *l* = floor (*N/t*), *t* = 1, 2, …, *N*, *R* is range series, *S* standard deviation series. Hurst exponent is defined as the slope of the least-squares regression line going through a cloud of partial time series [46].

Approximate Entropy (ApEn) is approximately equal to the negative mean natural logarithm of the conditional probability that two sequences that are similar at *M* points remain similar at the next point within tolerance r [47]. ApEn can be calculated as follows:

(8)$$ApEn=\frac{1}{N-M+1}\overset{N-M+1}{\underset{i=1}{\sum }}\ln C_{i}^{M}\left(r\right)-\frac{1}{N-M}\overset{N-M}{\underset{i=1}{\sum }}\ln C_{i}^{M+1}\left(r\right)$$
where $C_{i}^{M}$ is a measure of the regularity or frequency of similar time series sequences within a window of length M.

Sample Entropy (SampEn) is very similar to calculation ApEn. SampEn is also the negative natural logarithm of the conditional probability that two sequences similar for *M* points remain similar at the next point but it does not count self-matches [47].Decorrelation time is defined as the time of the first zero crossing of the autocorrelation sequence of a signal. If the decorrelation time is lower, the signal is less correlated [48].Hjorth parameters, such as Hjorth mobility (*HM*) and Hjorth complexity (*HC*) are computed from the power spectrum of the data. The mobility is the variance of the signal by the variance of the amplitude distribution of the time series:

(9)$$HM=\frac{1}{P}\overset{N/2}{\underset{k=1}{\sum }}p_{k}k^{2}$$
where $\left\{p_{k}\right\}=\left\{|s_{k}{|}^{2}\right\}$ is the power spectrum, $\left\{s_{k}\right\}$ is Fourier transform with k=1, … , N/2 for any frequency $f=f_{s}k/N$, $f_{s}$ is the sampling frequency, P is the total power of the signal.

The complexity is the variance of the rate of slope changes with reference to an ideal sine curve:(10)$$HC=\frac{1}{P}\overset{N/2}{\underset{k=1}{\sum }}p_{k}k^{4}$$

Hjorth parameters such as mobility and complexity can be also calculated in the time domain [49]. Mobility is defined as the square root of the ratio of the first derivative of the signal to the activity of the original signal:


(11)
$$M=\frac{\sigma_{x^{\prime }}}{\sigma_{x}}$$


Complexity is defined as the ratio of mobility of the first derivative of the signal to the mobility of the signal itself:(12)$$C=\frac{\sigma_{x^{\prime \prime }}/\sigma_{x^{\prime }}}{\sigma_{x^{\prime }}/\sigma_{x}}$$

Higuchi Fractal Dimension (*H_FD*) is the measure of complexity in the time domain (considering time series as a geometric object) [50]. The Higuchi fractal dimension can be expressed as following power law:(13)$$\langle L\left(T\right)\rangle \propto T^{-H\_FD}$$
where T is the time interval, 〈L(T)〉 is the average value over T of the curve length [51].

Katz Fractal Dimension is another method of dimension calculation which is derived directly from the signal [50]. Katz fractal dimension can be expressed as:

(14)$$D=\frac{\log_{10}n}{\log_{10}\left(N\right)+\log_{10}n}$$
where n=L/a is the number of steps in the curve, *a* is the average distance between successive points, N is the total length of the curve and d=max(distance(1,i)) is the distance between the first point of the signal and the point of the signal that provides the farthest distance.

Line length [52] can be derived from the fractal dimension normalization by Katz, defined as:

(15)$$FD=\frac{\log_{10}N}{\log_{10}\frac{d}{L}+\log_{10}N}$$
where *d* is the diameter estimated as the distance between the first point and the point for which the furthest distance can be obtained, and *L* is the total length of the curve or sum of distances between successive points, computed as:(16)$$L=\overset{N}{\underset{i=1}{\sum }}\left|x\left(i-1\right)-x\left(i\right)\right|$$

Signals can be generated by the so-called 1/*f* process. Such signals have a power law type relationship of the form:

(17)$$S_{x}\left(f\right)=\frac{constant}{{\left|f\right|}^{\gamma }}$$
where $S_{x}\left(f\right)$ is the power spectral density, f is the frequency and γ is spectral parameter which is usually close to 1 [53]. Linear regression of the Power Spectral Density (PSD) in the log–log scale can be used to estimate the slope and the intercept. The four characteristics are returned: intercept, slope, mean square error, and R2 coefficient.

Spectral Entropy is the measure of the Shannon entropy using the power spectrum of the signal [54]. Shannon entropy is used to capture the “peakiness” of probability mass function as follows:

(18)$$SpectEn=-\overset{N}{\underset{i=1}{\sum }}f_{i}\log_{2}f_{i}$$
where f is calculated by dividing the individual frequency components of the spectrum by the sum of all the components, *N* is the number of points in the spectrum.

Entropy of the singular value decomposition is a measure of dynamic complexity [55] which can be defined as:

(19)$$SvdEn=-\overset{N}{\underset{i=1}{\sum }}\frac{\lambda_{i}}{\sum_{i}\lambda_{i}}\log\frac{\lambda_{i}}{\sum_{i}\lambda_{i}}$$
where $\lambda_{i}$ is *i*-th eigenvalue of the matrix obtained from singular value decomposition.

The Fisher information [55] can be also defined using singular value decomposition as follows:


(20)
$$FI=\overset{N}{\underset{i=1}{\sum }}\frac{{\left(\frac{\lambda_{i+1}}{\sum_{i}\lambda_{i}}-\frac{\lambda_{i}}{\sum_{i}\lambda_{i}}\right)}^{2}}{\frac{\lambda_{i}}{\sum_{i}\lambda_{i}}}$$


Wavelet transform decomposes signal to the time–frequency domain at different decomposition levels [39]. The result of a wavelet transform can be complex wavelet coefficients. The modules of these complex wavelet coefficients represent the energy of the original signal at different decomposition levels occurring at different times.Wavelet transform combined with a Teager–Kaiser energy operator can effectively separate local amplitude and frequency fluctuations [56]. Teager–Kaiser energy operator can be defined as follows:

(21)$$\varphi \left(x_{i}\right)=x_{i}^{2}-x_{i-1}x_{i+1}$$
where $x_{i}$ is a discrete signal. Calculating the energy of a signal requires only three samples. Hence, the energy operator has a small time window making it ideal for local (time-based) signal analysis.

Permutation entropy is a complexity measure based on comparison of neighboring values in signal [57]. The permutation entropy can be calculated follows:

(22)PermEn=−∑p(π)logp(π)
where the sum runs over all *n*! permutations *π* of order *n*.

Detrended fluctuation analysis (*DFA*) is a measure for quantifying long-range correlations of data [58]. The root mean square fluctuation integrated and detrended signals can be calculated as:

(23)$$DFA\left(M\right)=\sqrt{\frac{1}{N}\overset{N}{\underset{i=1}{\sum }}{\left(y_{i}-y_{i}\left(M\right)\right)}^{2}}$$
where $y_{i}=\sum \left(x_{i}-\bar{x_{i}}\right)$, *M*—length of the boxes, $y_{i}\left(M\right)$ means the y coordinate of the straight line segment (detrend).

Lempel–Ziv complexity is a distance measure based on the relative information [59,60]. Normalized Lempel–Ziv complexity can be defined follows:

(24)$$C_{LZ}=N\cdot c\left(N\right)\log_{2}N$$
where c(i) means the number of distinct patterns of sequence $x_{i}$.

Features determined on the basis of the parameters presented above were divided into 7 groups:Group I consists of the mean, variance, standard deviation of the data, peak-to-peak amplitude, skewness, kurtosis and Hurst exponent;Group II consists of approximate entropy, Sample Entropy, permutation_entropy, detrended fluctuation analysis, spectral entropy, Lempel–Ziv complexity, singular value decomposition entropy, SVD Fisher Information;Group III consists of Hjorth parameters such as Hjorth mobility and Hjorth complexity computed both in the time domain and from the power spectrum of the data;Group IV concerns fractal parameters and encompasses Higuchi Fractal Dimension, Katz Fractal Dimension and line length calculated based on Katz Fractal Dimension;Group V consists of Spectral Entropy as well as features calculated employing linear regression of the Power Spectral Density (PSD) in the log–log scale: intercept, slope, mean square error and R2 coefficient;Group VI incorporates the six energy of wavelet decomposition coefficients;Group VII consists of fourteen Teager–Kaiser energy parameters.

Since the values of obtained parameters vary significantly from one another it becomes necessary to standardize them before classification using the following equation:(25)$$x_{std}=\frac{x_{old}-\bar{x_{old}}}{\sigma }$$
where:

σ—standard deviation of the *i*-th feature value before standarization;

$\bar{x_{old}}\;$—mean of the *i*-th feature value before standarization.

### 3.3. Base Classifiers

An important role within the process of designing ensemble classifiers is held by base classifiers. Within the presented solution it had been decided to test several well-known algorithms. Since one of the fundamental assumptions is the necessity to recognize a particular subject within a fraction of a second as well as that training/retraining needs to happen within no more than several minutes, the use of classifiers such as support vector machines, radial basis networks, and deep neural networks was abandoned. Every base classifier was trained separately for every group of attributes listed in Section 3.2 using 10 folds of cross-validation. Each time the same division of data into folds was utilized thanks to which results obtained by different base classifiers were comparable.

#### 3.3.1. *k* Nearest Neighbor (kNN)

The *k* nearest neighbor classifier (kNN) is one of the simplest classifiers. It decides to assign a new point in feature space to a particular class based on distances from that point to its *k* nearest neighbors. This distance is treated as the inverse of the probability measure. The distance may be determined using various metrics, among which the most popular include the Euclidian distance, city (Manhattan) metric, or the Chebychev distance. This is a lazy type classifier, so the model creation time is equal to 0.

#### 3.3.2. Naive Bayes

The Naive Bayes classifier is based on the Bayes Theorem:(26)$$P(C|X)=\frac{P(X|C)\;P\left(C\right)}{P\left(X\right)}$$
where:

*P*(*C*)—the prior probability of class *C*;

*P*(*X*|*C*)—the likelihood which is the probability of predictor *X* given class *C*;

*P*(*X*)—is the prior probability of predictor.

In this classifier it is assumed that each input variable is independent, which is usually not true with respect to real data. Hence the word naive in the name. However, despite this unrealistic assumption Bayes classifiers often produce good results.

#### 3.3.3. Artificial Neural Networks (ANN)

Artificial neural networks are the best-known and most often utilized method of artificial intelligence. ANNs consist of appropriately connected structures consisting of single artificial neurons. The model of an artificial neuron is only roughly based on the manner and construction of a real neuron. In ANNs, signals provided to the input nodes are multiplied by values called weights connected to individual synaptic connections. The processing of information also occurs within the neuron itself. The training of an artificial neuron network comes down to the selection of weights in a way that an answer (output) of the ANN for a provided input signal is as close as possible to the desired value. Within the present work feedforward networks (MLPs) with no more than two hidden layers were used.

#### 3.3.4. Classification and Regression Trees

Classification and Regression Trees (CART) are binary trees (there are only two branches from each node) with one-dimensional divisions. Within the node of the tree, a condition is created by verifying all possible divisions in points that are mid-points of segments between subsequent sorted *x_j_* and *x_j_*_+1_ values. The best division is one that separates input data into relatively homogeneous subsets. Impurity assessment *I*(*tr*) after the division may be done, for example, by applying Gini’s index:(27)$$I\left(tr\right)=1-\overset{N_{c}}{\underset{j=1}{\sum }}p_{j}^{2}$$
where:

*p_k_*—frequency of the occurrence of elements from class *j* after the division;

*N_c_*—the number of all classes.

#### 3.3.5. Linear Discriminant Analysis

Linear discriminant analysis (LDA) finds a linear combination of features that best differentiate between classes. Combinations of results are used as linear classifiers or to reduce the dimensionality of the input space. The present work made use of the regularized linear discriminant analysis described in detail in [61], where it is assumed that all classes possess the same covariance matrix:(28)$$\hat{\Sigma_{\gamma }}=\left(1-\gamma \right)\hat{\Sigma }+\gamma diag\left(\hat{\Sigma }\right)$$
where:

$\hat{\Sigma }$ is the empirical, pooled covariance matrix;

*γ* is the amount of regularization.

### 3.4. Ensemble Classifiers

The recognition of people comes down to the issue of classification where the number of classes is equal to the number of people present in the database (people who, for example, have access to resources). Within the present work two techniques for combining classifier decisions were utilized:-majority vote;-weighted vote with weight based on rank order.

The authors are aware of the existence of several other methods for the combining of base classifier decisions; however, a choice to use two relatively simple methods which most likely will result in underestimated results of classification has been made.

The first of these methods assumes that a classifier decision is equal to the class which has been indicated by most base classifiers. In an event when two or more classes have been expressed by the same number of base classifiers, then the decision receives a “NONE” label and no recognition is possible. This naturally increases the false rejection rate.

The decision of the entire set of classifiers made on the basis of a weighted vote with weights based on rank order assumes the acceptance of particular weights. The weighted value connected to every label depends on rank *R*, which has been determined based on the accuracy of all base classifiers The final decision was the class label with the largest total of weights:(29)$$cl=\arg\max\left(\overset{k}{\underset{j=1}{\sum }}w_{j}\cdot d_{j,i}\right)$$
where: *cl*—class label; *k*—the number of base classifier (seven in this paper), *w_j_* = [*w*_1_, …, *w_R_*, …, *w_k_*]—weights, which are calculated from the following formula:(30)$$w_{R}=\frac{k+1-R}{k}$$
where: *R*—indicates the rank for *j*-th classifier, *R* = {1, 2, …, *k*}. *d_j,i_*—decision (class)of the *j*-th classifier, *d_j,i_* ∈ {0, 1}. If *j*-th classifier chooses class *i* then *d_j,i_* = 1 otherwise *d_j,i_* = 0.

It was accepted that a person is unrecognized (which meant that the person was not in the database) if at least two classes had the same total weight or if the final total was smaller than the arbitrarily chosen threshold *Th*. In those cases, the person was given a ‘NONE’ label. The accepted threshold permits a minimum required level of similarity to consider the person being scrutinized as identified.

### 3.5. The Study Group

The research was carried out at the laboratories of the Faculty of Mechanical Engineering of the Bialystok University of Technology, on a sample of 322 people including 139 women and 183 men. All study participants were informed about the aims and manner of conducted tests and signed appropriate declarations. Participants were free to withdraw from the study at any time for any reason. During the tests, at a sign from the researcher, participants walked through a testing path whose length exceeded 10 m and within which there were hidden two force plates manufactured by the Kistler Company. The participants were not informed about the presence or the location of the plates nor about having to step on one. In cases where the participant did not step on the plate or its edge the measurement was conducted again, but with a modification to the person’s starting point with respect to the previous attempt. Every participant walked in their own sports shoes and at a speed of their choosing. During the experiment, after every 10 gait strides with a single person, there was a short, 1–2 min, break to prevent the subject from becoming tired. A range of 14 to 20 gait cycles were carried out with every participant. A total of 5980 gait cycles were recorded.

## 4. Results

Table 1, Table 2, Table 3, Table 4, Table 5, Table 6, Table 7 and Table 8 contain the results of the accuracy of person recognition expressed using the Correct Classification Rate. Additionally, the time needed for model creation for all 10 folds (*T_M_*), as well as its operation during the recognition process of individual participants (*T_R_*), was also included. Both values include the time needed to determine the parameter values within a given group of data. Times were determined using a computer containing an Intel Core i7-9750H, 2.6 Ghz processor with all other applications turned off. To minimize the impact of system processes on calculation results they were conducted 10 times. The tables presented below include the mean and standard deviation of these results. As has been mentioned before, the *k* nearest neighbor’s classifier is a lazy type of classifier, which means that it does not require to construct a model, therefore, the *T_M_* value was left blank (‘-’) for this algorithm. It must be said, however, that the presented times do not include times of feature calculation. Times for the calculation of individual groups of parameters are contained in Table 1. This approach was used because the times to determine parameters are much larger and would obscure the differences in the activity of individual classifiers.

Results presented for individual base classifiers assume function with the use of parameters of providing the best results. The study, of course, was conducted for numerous settings of selected classifiers. Concerning the *k*NN classifiers, the best results were seen with the Manhattan metric and the number of neighbors between 3 and 7, depending on the group of parameters. During the selection of the best *k*NN classifier parameters, it quickly became apparent that the choice of a given distance had a decidedly greater impact on obtained results than the number of neighbors. For CART, better results were seen for small values of minimal leaf size. When using artificial neuron networks, on the other hand, the number of accurate classifications grew after the neuron activation function was changed to a rectifier linear unit. It must be said, however, that the utilization of parameters different from those presented below has a lesser impact on results than the type of selected classifier. The row in bold in Table 2, Table 3, Table 4, Table 5, Table 6, Table 7 and Table 8 shows the best classification results obtained within a given group of data. Additionally, the classification results (CCR) are presented graphically in Figure 3.

**Table 2 sensors-23-00508-t002:** Classification results for individual classifiers with Group 1 attributes.

Type of Classifier	CCR (%)	*T_M_* (s)	*T_R_* (µs)
kNN	96.76	-	45.113 ± 0.900
Naive Bayes	96.07	4.861 ± 0.016	1928.350 ± 10.05
CART	72.66	19.917 ± 0.115	8.462 ± 0.436
MLP	93.14	589.174 ± 11.82	9.313 ± 0.179
**rLDA**	**99.46**	**221.789 ± 0.719**	**61.954 ± 1.281**

**Table 3 sensors-23-00508-t003:** Classification results for individual classifiers with Group 2 attributes.

Type of Classifier	CCR (%)	*T_M_* (s)	*T_R_* (µs)
kNN	94.46	-	69.712 ± 2.326
Naive Bayes	90.85	7.486 ± 0.111	3245.350 ± 20.42
CART	52.51	40.589 ± 0.281	8.629 ± 0.313
MLP	86.54	433.859 ± 37.79	10.139 ± 0.242
**rLDA**	**98.03**	**236.266 ± 5.298**	**85.425 ± 7.356**

**Table 4 sensors-23-00508-t004:** Classification results for individual classifiers with Group 3 attributes.

Type of Classifier	CCR (%)	*T_M_* (s)	*T_R_* (µs)
**kNN**	**92.41**	**-**	**33.947 ± 0.932**
Naive Bayes	91.40	3.351 ± 0.029	1090.342 ± 16.35
CART	58.88	13.190 ± 0.478	9.782 ± 0.759
MLP	89.01	534.973 ± 29.13	10.335 ± 0.232
rLDA	91.49	223.452 ± 2.265	143.036 ± 6.370

**Table 5 sensors-23-00508-t005:** Classification results for individual classifiers with Group 4 attributes.

Type of Classifier	CCR (%)	*T_M_* (s)	*T_R_* (µs)
kNN	88.24	-	34.876 ± 1.194
Naive Bayes	89.60	3.196 ± 0.251	852.916 ± 24.24
CART	48.61	10.993 ± 0.166	10.142 ± 4.202
MLP	85.23	444.741 ± 26.56	10.354 ± 0.247
**rLDA**	**93.73**	**227.074 ± 2.68**	**132.890 ± 3.400**

**Table 6 sensors-23-00508-t006:** Classification results for individual classifiers with Group 5 attributes.

Type of Classifier	CCR (%)	*T_M_* (s)	*T_R_* (µs)
kNN	82.78	-	238.655 ± 9.141
Naive Bayes	78.73	3.997 ± 0.099	1412.096 ± 33.66
CART	37.26	18.425 ± 0.386	9.368 ± 0.384
MLP	74.89	230.279 ± 12.87	10.220 ± 2.142
**rLDA**	**92.59**	**221.166 ± 5.867**	**54.590 ± 4.280**

**Table 7 sensors-23-00508-t007:** Classification results for individual classifiers with Group 6 attributes.

Type of Classifier	CCR (%)	*T_M_* (s)	*T_R_* (µs)
kNN	77.56	-	59.199 ± 10.17
**Naive Bayes**	**83.58**	**4.285 ± 0.134**	**1631.191 ± 10.59**
CART	35.90	23.629 ± 0.428	9.270 ± 0.207
MLP	69.90	580.153 ± 7.431	9.509 ± 0.205
rLDA	70.48	222.276 ± 3.009	110.673 ± 12.55

**Table 8 sensors-23-00508-t008:** Classification results for individual classifiers with Group 7 attributes.

Type of Classifier	CCR (%)	*T_M_* (s)	*T_R_* (µs)
kNN	92.11	-	88.946 ± 3.242
Naive Bayes	91.82	8.813 ± 0.0546	3789.095 ± 53.99
CART	55.75	51.430 ± 0.851	8.961 ± 0.369
MLP	88.11	582.196 ± 35.290	9.432 ± 0.319
**rLDA**	**96.05**	**215.444 ± 3.0468**	**186.881 ± 16.58**

The result analysis of selected classifiers shows that there does not exist one type of classifier which would allow the highest level of accurate recognition for all data groups. Generally, very good results were obtained using the *k*NN classifier. Its major advantage is the fact that model creation is not necessary. Even though this extends the operation time needed for the recognition of a particular person, a comparison of data from the *T_R_* column indicates that this time is lower than 0.25 ms (Table 6), which is fully acceptable. The *k*NN classifier allowed the best classification results for attributes from Group 3 (Table 4). These results were better than those obtained by the next best type of classifier by nearly 1 percentage point.

The Naive Bayes classifier had the shortest *T_M_* model construction time (excluding the *k*NN classifier). At worst, this time never exceeded 8 s. The Naive Bayes classifier presented quite good classification results, and in recognizing people based on characteristics contained within group 6 these were the best (Table 7). It is worth stating that the next classifier that worked on this data produced results that were more than 6 percentage points worse than the Naive Bayes. One inconvenience connected to this type of classifier is its longest pattern recognition time (*T_R_*), which was from 6.5 (Table 5) to nearly 35 times (Table 6) longer than that of the results of the next-best type of classifier. It must be noted, however, that the time needed to recognize a person that is less than 4 ms is fully acceptable.

Regardless of the data group the CART classifier gave the worst results when compared to the other selected classifiers. Only the results concerning Group1 parameters (Table 2) can be considered somewhat decent. Since in many cases its CCR was below 50% this type of classifier was abandoned in further studies. The negative assessment of this classifier was not even impacted by the fact that its recognition time (*T_R_*) was the lowest of all types of considered classifiers.

A trait that is characteristic of feedforward neural networks is their long training time. Regardless of which group of data was used, its *T_R_* model creation time was the longest. At times it approached the 10-min mark and at best it was 58 times longer than model creation using the Naive Bayes. What is more, this long training time did not translate into better recognition results. The best results were not obtained with any of the data groups. At all times the MLP classifier was only better than CART.

The rLDA classifier is the best of the considered classifiers. It generated the best results for 5 of the 7 groups of features. If any other classifier presented better results, then the rLDA was second (Table 2) or third (Table 7). It must be stressed that the results for Group1 parameters are as high as 99.46% of accurate recognitions, which is one of the best results of human gait recognition based on GRFs signals reported in literature. One inconvenience resulting from the use of the rLDA classifier is a relatively long model creation time, which is between 3.5 and 4 min, as well as a relatively long time of recognition lasting 0.1 ms. These times, however, fall within accepted assumptions and are more than made up for by its very good recognition rates.

Based on obtained results, an ensemble of classifiers consisting of base classifiers that have shown the best recognition results for individual groups of features (Table 2, Table 3, Table 4, Table 5, Table 6, Table 7 and Table 8) was created. Subsequently, a set of heterogeneous classifiers consisting of 5 rLDA classifiers, as well as one *k*NN classifier (Group 4) and one Naive Bayes classifier (Group), was compiled. Additionally, to compare results, 4 sets of homogeneous classifiers where base classifiers included *k*NN, Naive Bayes, MLP and rLDA were also assembled. Classification results for majority vote (Table 9) and weighted vote (Table 10) have been presented below.

Weighted vote with weight based on rank order provides higher recognition rates by approximately 0.5 percentage points than regular majority vote recognition. In all cases, except for the ensemble of homogeneous classifiers based on rLDA, majority vote presented better results than any individual classifiers of a given type. It is also worth noting that the greatest improvement in quality concerning a set of classifiers in comparison with a single classifier occurred with MLP and the smallest happened with rLDA.

Classification time by an ensemble of classifiers depended, with the exception of MLP, mainly on base classifier classification time and, more precisely, on the slowest of them. Decision time by a set of classifiers is dependent on algorithm type. It has been noted that weighted vote with weight based on rank order took a bit less time than majority vote, but it was characterized by greater variability.

Results obtained using weighted vote with weights based on rank order reached 99.65% of accurately recognized patterns. These results exceed all others obtained through the utilization of different methods. A comparison of values contained within Table 8, Table 9 and Table 10 shows that heterogeneous ensembles of classifiers are better than homogeneous sets. One flaw of heterogenous classifiers is recognition time, which is significantly impacted by the operation time of the slowest of base classifiers. As has been mentioned earlier, the times presented above do not include the time needed to determine feature values for individual strides (Table 1). When values from Table 1 and Table 11 are compared, it is easy to see that the decision time of the set of classifiers along with that needed to determine feature values will be less than 89 ms. In turn, classifier training time with a database containing 5980 strides, like the one used for the present paper, will amount to less than 12.5 min.

## 5. Discussion

Results presented within the present work were obtained using GRF database, which is the largest with respect to the number of people tested and one of the largest in relation to the number of recorded strides [40,42]. It is also worth mentioning that the collected research material was obtained on a relatively homogeneous group of people, which makes the difficulty level of classifying these people higher. Despite all of that, the obtained results were very good, and it is one of the best results when compared to those available in literature. A slightly higher level of recognition, that of 99.8%, had been obtained by [33]. However, the authors of those tests conducted them on a sample group of only 50 people which, as has been shown by [43], significantly impacts the results obtained. Equally high recognition rates have been presented in [36], where a method called Gait-ViT allowed the accurate recognition of 99.51 to 100 percent of gait cycles. In the work of [36], in contrast to [33], very good results were also gained for a database containing a larger number of people (OU-LP data set, 3916 participants). It should be noted that the difficulty in the direct comparison of results lies in the fact that in both of the above-mentioned works, images from a video camera were used as data describing human gait.

When it comes to works utilizing measurements connected with FS, results obtained within the present work are even better. Source [62] proposed a special version of a convolutional-recurrent neural network (KineticNet) which, based on vertical, anterior/posterior of GRFs as well as one of the coordinates of the center of pressure, was able to re-identify 118 subjects with approximately 96% accuracy. Similar results were obtained by [63] where an accuracy of 94% was observed using Fine Gaussian Support Vector Machines and GRFs. The authors, unusually, determined training time to be 3.312 s but the research concerned only 5 people. In [39] a kNN group of classifiers was used where the results obtained reached a level of >95% on a sample of only 10 people. Less favorable results than those presented within this work have been shown by [42], where for 127 people the EER ranged from 2.5% to 10% depending on the experiment setup. In [41] (UbiFloorll and MLP) the error amounted to 1% where the sample group consisted of 10 people. In Ref. [64], in turn, which examined foot pressure patterns, the error was in best cases, 0.6% to 7.1% for 104 people and 520 strides. These results are even better than in previous works of the first author of this work [22], where a result of just below 99% of accurate recognitions on a group of 220 people was obtained.

## 6. Conclusions

Results of the recognition of people based on ground reaction forces during walking show the great potential of human gait as a biometric. Obtained results confirm that heterogeneous complex classifiers exceed in quality all other considered simple classifiers. The utilization of various types of classifiers as base classifiers allows the achievement of slightly better results than when homogeneous classifiers are used. Further work in this area can be carried out in two directions. Firstly, more advanced methods of combining the decisions of individual classifiers need to be tested, and their impact on the final outcome needs to be assessed. Secondly, it is necessary to verify the resistance of a set of heterogeneous classifiers to changes in the movement patterns of people undergoing the procedure of recognition caused, for example, by different footwear as well as symmetric or asymmetric loading.

## Figures and Tables

**Figure 1 sensors-23-00508-f001:**
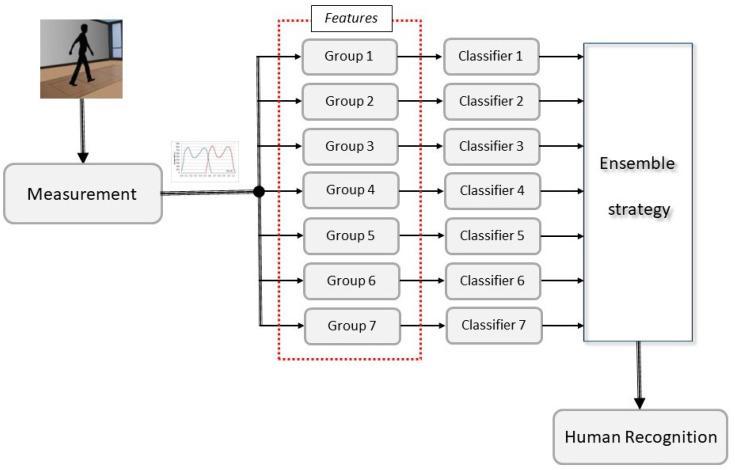
The overall flowchart of the proposed idea.

**Figure 2 sensors-23-00508-f002:**
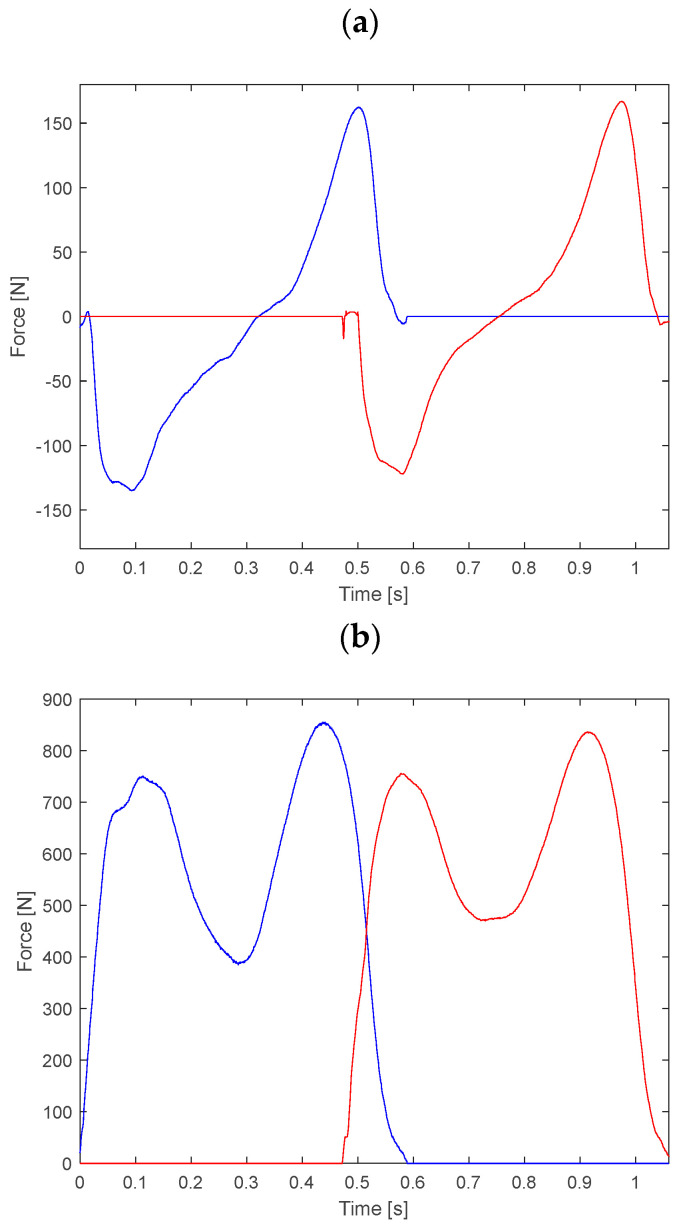
Components of GRF in: (**a**) anterior/posterior—Fx; (**b**) vertical—Fy; (**c**) medial/lateral—Fz direction of the left lower limb (blue line) and of the right one (red line) in sport shoes.

**Figure 3 sensors-23-00508-f003:**
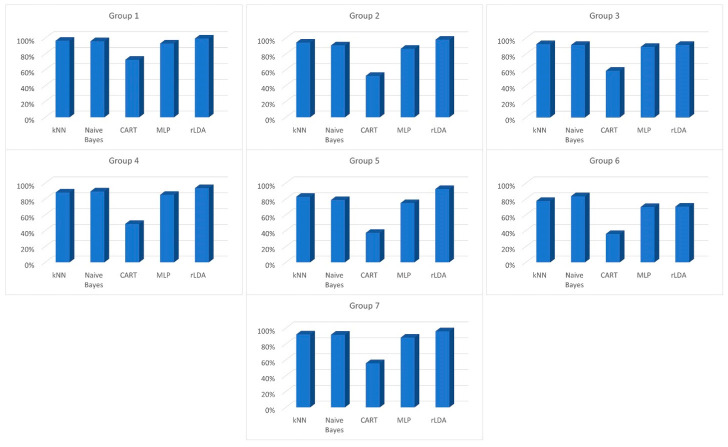
Correct Classification Rate (%) of base classifiers for each group of features.

**Table 1 sensors-23-00508-t001:** Average time and standard deviation for determining parameter values for one gait cycle depending on parameter group.

Set of Features	Group 1	Group 2	Group 3	Group 4	Group 5	Group 6	Group 7
Average time (ms)	17.543	87.373	3.662	0.426	4.031	0.806	1.419
SD	0.236	1.316	0.087	0.009	0.116	0.050	0.058

**Table 9 sensors-23-00508-t009:** Classification results for ensemble of homogeneous classifiers, majority vote.

Type	CCR (%)	FRR (%)	FAR (%)	*T_R_* (µs)
kNNs	98.71	0.62	0.67	283.244 ± 5.324
Naive Bayes	97.93	1.10	0.97	3295.642 ± 15.32
MLP	97.99	1.46	0.55	57.842 ± 2.942
rLDA	99.13	0.49	0.38	204.923 ± 13.54

**Table 10 sensors-23-00508-t010:** Classification results for ensemble of homogeneous classifiers, weighted vote.

Type	CCR (%)	FRR (%)	FAR (%)	*T_R_* (µs)
kNNs	98.85	0.12	1.03	274.632 ± 7.325
Naive Bayes	98.46	0.25	1.29	3282.942 ± 17.37
MLP	98.41	0.17	1.42	49.034 ± 4.332
rLDA	99.55	0.08	0.37	198.422 ± 17.14

**Table 11 sensors-23-00508-t011:** Classification results for ensemble of heterogeneous classifiers.

Type of Voting	CCR (%)	FRR (%)	FAR (%)	*T_R_* (µs)
Majority voting	99.30	0.52	0.18	1683.049 ± 12.72
Weighted voting	99.65	0.03	0.32	1667.927 ± 13.93

## Data Availability

Not applicable.

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
