# Peer review of "Ensemble of Heterogeneous Base Classifiers for Human Gait Recognition"

_sensors, 2023, doi:10.3390/s23010508_

Round 1
Reviewer 1 Report
Based on the traditional methods of human gait recognition, this paper proposes an integrated approach based on GRFs and an ensemble of heterogeneous base classifiers. The detection accuracy is improved compared with other methods. It is an excellent article with practical significance, but there are still some things that need to be improved. Please correct the following problems:
1. In the abstract, it is recommended that you explain the feasibility of using heterogeneous base classifier ensembles so that the reader can better understand your paper.
2. Your introduction could have been more straightforward. Firstly, the introduction logic of the human gait recognition method could be better. You can start with the classification of the current process and then introduce the pros and cons of the present scenario. Still, you submit the method category in the middle of explaining the pros and cons of the current approach.
3. In addition, some contents in the paper are very abrupt, such as the paragraph quoted in literature 30-33. Before the section is the type of the method after the paragraph is the introduction of the technique. It is inappropriate for this paragraph to appear here. In the end, you say that you have come up with a method, but you have not thought of the advantages of this method.
4. “3.2. Features” appear in the second chapter, which is wrong. Please revise it. The paper 10.1108/AA-12-2021-0174 provides some suggestions.
5. In terms of the experiment, the contribution of this paper is modest because it simply integrates several traditional methods and does not propose new ideas to improve accuracy or shorten detection time.
6. Pay attention to the uniform formatting of the paper. For example, the format of the references should be the same. Moreover, please review the article again and correct the minor errors.
Reviewer 2 Report
This work uses an ensemble of heterogeneous base classifiers to address the issues of biometric systems. A group of parameters identified on the basis of ground reaction forces was accepted as input signals. The proposed solution was tested on a sample of 322 people. Results concerning the accuracy of recognition meaning the Correct 19 Classification Rate quality at 99.65% as well as operation time: the time of model construction at <12.5 minutes and the time needed to recognize a person at <0.1 seconds should be considered as very good and exceed in quality other methods so far described in literature. The authors have addressed a good problem, however, the following major concerns need to be addressed before the paper can be accepted for the publication:
1- Please incorporate subsections motivation and contributions within introduction section of the paper.
2- Please provide a separate section as related work in the paper. Additionally, some recent works related to image and video classification need to be introduced within related work section of the paper. Some suggested papers are given below for the same:
Robust hand gestures recognition using a deep CNN and thermal images
3- Either provide the complete block diagram of the proposed method or provide the algorithmic development of the proposed method which would be helpful for the reader.
4- Results discussions need to be elaborated for both, figures and tables incorporated in the paper.
5- Please correct the equations labelling. Equation number must come in front of the equation rather than below of the same.
Overall, presented work is addressing a good problem, however, the aforementioned suggestions need to be incorporated in the revised manuscript.
Round 2
Reviewer 2 Report
The authors have addressed all my concerns. The paper can be accepted in it's current form.